# Evaluation of culture- and PCR-based methods for detecting *Burkholderia pseudomallei* in soil samples in Thailand

**Didtawa Suttisak[1], Charlene Mae Salao Cagape[1], Rathanin Seng[2], Natnaree Saiprom[1], Wongwarut Boonyanugomol[2], Kamolchanok Rukseree[2], Paul J. Brett[1,3], Mary N. Burtnick[1,3], Narisara Chantratita** [1,4]*

**1** Department of Microbiology and Immunology, Faculty of Tropical Medicine, Mahidol University, Bangkok, Thailand, **2** Department of Medical Science, Amnatcharoen Campus, Mahidol University, Amnat Charoen, Thailand, **3** Department of Microbiology and Immunology, University of Nevada, Reno School of Medicine, Reno, Nevada, United States of America, **4** Mahidol-Oxford Tropical Medicine Research Unit, Faculty of Tropical Medicine, Mahidol University, Bangkok, Thailand

* narisara@tropmedres.ac

## Abstract

### Background

*Burkholderia pseudomallei* is an environmental bacterium that causes melioidosis, a life-threatening disease prevalent in tropical regions. Accurate detection of *B. pseudomallei* in the environment is essential for identifying areas that pose an infection risk. This study aimed to evaluate culture- and real-time polymerase chain reaction (PCR)-based methods for detecting *B. pseudomallei* directly in soil samples, and to assess its environmental prevalence in relation to antibody levels in healthy individuals from Amnat Charoen, Northeast Thailand.

### Methodology/principal findings

Initial studies used spiked soil samples to evaluate culture-based methods including Ashdown agar, phosphate-buffered acidic erythritol (ACER) agar, Threonine-basal salt solution with colistin 50 (TBSS-C50) broth, TBSS-C50 based erythritol broth, and ACER broth, and real-time PCR assays targeting *BPSS1187* and *TTS1-orf2*. Hemolysin coregulated protein 1 (Hcp1)-specific antibodies were measured in 398 plasma samples using an enzyme-linked immunosorbent assay. Based on these antibody titers, 238 soil samples were collected from the households of individuals that were strongly positive (N = 6) and negative (N = 6) for Hcp1-specific antibodies. *B. pseudomallei* was isolated from 3.36% (8/238) of soil samples by direct culture on Ashdown agar and from 5.46% (13/238) after enrichment in ACER broth followed by culture on Ashdown agar. Real-time PCR assays confirmed the presence of *B. pseudomallei* in 2 out of 20 culture-positive soil samples. Overall, *B. pseudomallei* was detected in 14

**Data availability statement:** All relevant data are within the manuscript and its Supporting Information files.

**Funding:** o This work was supported by the Defense Threat Reduction Agency (DTRA), USA [HDTRA1-21-C-0014 to PJB, MNB, NC] and the Wellcome Trust [220211 to NC]. For the purpose of Open Access, the author has applied a CC BY public copyright license to any Author Accepted Manuscript version arising from this submission. The funders had no role in study design, data collection and analysis, decision to publish, or preparation of the manuscript.

**Competing interests:** The authors have declared that no competing interests exist.

of 129 samples (10.9%) from households of seropositive individuals and in 6 of 108 samples (5.5%) from households of seronegative individuals ($P = 0.165$).

## Conclusions/significance

This study demonstrates the importance of using effective methods for detection of *B. pseudomallei* in environmental samples. For culture-based approaches, enrichment in ACER broth followed by subculture on Ashdown agar demonstrated enhanced bacterial recovery compared to direct culture on Ashdown agar alone. There was limited detection of *B. pseudomallei* in soil by real-time PCR, and lack of an association between environmental detection methods and human seroprevalence. These findings highlight the need for integrated, multi-dimensional surveillance approaches.

### Author summary

Melioidosis is an infectious disease caused by the environmental bacterium *B. pseudomallei,* which is found in soil and water in tropical regions. Detection of *B. pseudomallei* in the environment is essential for identifying high-risk areas for melioidosis and for raising clinical awareness, particularly in endemic regions. This study evaluated culture- and real-time PCR-based methods for direct detection of *B. pseudomallei* in soil samples and assessed the environmental prevalence of *B. pseudomallei* in relation to antibody levels among healthy individuals in Amnat Charoen Province, Northeastern Thailand. The findings provide important insights into the association between environmental prevalence of *B. pseudomallei* and seroprevalence in the local population. Moreover, the study identifies the effective methods for environmental detection of *B. pseudomallei*, which is critical for improving surveillance, guiding public health interventions, and enhancing clinical preparedness in areas at risk for melioidosis.

## Introduction

*Burkholderia pseudomallei* is a Gram-negative, saprophytic environmental bacterium that causes melioidosis, a potentially fatal infectious disease affecting humans and animals [1]. This pathogen is predominantly found in moist soils, surface and ground water, and the rhizosphere in tropical and subtropical regions [2]. *B. pseudomallei* infections typically occur through inoculation, ingestion, or inhalation of aerosolized bacteria, with most cases reported during the wet season or following severe weather events such as tropical storms [3]. Thailand reports approximately 7,000 cases of human melioidosis annually [4] with mortality rates of approximately 25–40% [5]. The clinical manifestations of melioidosis vary in severity, ranging from asymptomatic cases to localized skin infections, pneumonia, and septicemia [6,7]. In severe cases, the disease progresses rapidly, leading to acute septicemia, septic shock, and death. Consequently, prompt and accurate diagnosis, along with timely treatment, is crucial for effective disease control and prevention.

Culture-based methods remain the gold standard for detecting *B. pseudomallei* in clinical samples and are also recommended for environmental detection [8]. Ashdown agar, a selective medium, is effective for isolating *B. pseudomallei* in regions where the disease is endemic [9]. Traditionally, detection of *B. pseudomallei* in soil involved enrichment in threonine-basal salt solution with colistin (TBSS-C50), followed by subculture on Ashdown agar. However, this culture method has limited sensitivity, with detection rates of approximately 9.8% after 48 hours and 25.5% after 144 hours of enrichment [10]. Improved sensitivity has been achieved by varying nitrogen and carbon sources in TBSS-C50. For example, $NH_4H_2PO_4$ can be used as a substitute for nitrilotriacetic acid, and erythritol can replace threonine since it is a unique carbon source that is used by *B. pseudomallei* but not by other *Burkholderia* species [11]. Recently, a novel phosphate-buffered acidic erythritol (ACER) medium has been shown to enhance growth of *B. pseudomallei* compared to TBSS-C50 and TBSS-C50-based erythritol media in a study conducted in the north-central region of Vietnam [12]. Even with these improvements, culture methods remain limited by their low sensitivity and labor-intensive nature. To overcome these limitations, molecular methods such as real-time polymerase chain reaction (PCR) and Clustered Regularly Interspaced Short Palindromic Repeats (CRISPR) [13] have been employed for the detection of *B. pseudomallei* in environmental samples.

The current species-specific target used for the detection of *B. pseudomallei* by PCR is the *orf2* gene, which is located within the type III secretion system 1 (TTS1) gene cluster [14,15]. A real-time PCR assay targeting *TTS1-orf2* gene demonstrated 100% specificity in spiked human blood samples [15]. In a study using 200 soil samples collected from rice paddies in Northeast Thailand, *TTS1*-based real-time PCR showed the detection rate of 76.5%, and when *BPSS0745* was added as a secondary target the detection rate increased to 90% [16]. Real-time PCR has consistently shown higher detection rates for *B. pseudomallei* in environmental samples compared to culture methods using TBSS-C50 enrichment broth [17]. Recent advancements in real-time PCR assays for *B. pseudomallei* includes the incorporation of an internal control to ensure the efficiency of DNA extraction and the assessment of additional target genes including *BPSS0087, BPSS1187, BPSS1498* (*hcp1*), and *BPSS1492* (*bimA*) [18]. Among these, the assay targeting *BPSS1187* exhibited the highest sensitivity (76.3%) and specificity (100%) using plasma samples [18].

Optimizing methods for detection of *B. pseudomallei* is crucial for accurately mapping endemic areas and guiding surveillance efforts that could help to enhance awareness and inform disease prevention strategies. To determine the most effective methods for detecting *B. pseudomallei* in environmental samples, we initially evaluated both culture- and real-time PCR-based methods using spiked soil samples. We compared the performance of ACER agar with Ashdown agar, and evaluated ACER broth, TBSS-C50-based erythritol broth, and TBSS-C50 broth for culturing *B. pseudomallei* from spiked soil samples. We also determined the efficiency of real-time PCR assays targeting *BPSS1187* and *TTS1-orf2* using these samples. We then examined the environmental prevalence of *B. pseudomallei* in 238 soil samples collected in Amnat Charoen province. These included soil surrounding the houses of individuals that were strongly positive (N = 6) and negative (N = 6) for Hcp1-specific antibodies. Amnat Charoen borders Ubon Ratchathani, a recognized melioidosis endemic area, but currently lacks serological surveillance data.

## Materials and methods

### Ethics statement

The study was approved by the Ethics Committee of the Faculty of Tropical Medicine, Mahidol University (MUTM 2024-052-01) and the Mahidol University Central Institutional Review Board (MU-CIRB 2023/028.1503). Written informed consent was obtained from all participants or their parents or guardians prior to their enrollment. This study was approved by the Institutional Biosafety Committee of Mahidol University (MU 2024–010).

### Bacterial strains

*B. pseudomallei* isolates were obtained from both human and environmental sources in Northeast Thailand. The reference strain K96243 was isolated from a patient in Khon Kaen province [19]. Ten environmental isolates were obtained from soil

samples across different locations in Surin Province, Thailand, including Khwao Sinarin (strains 30–191-S08, 30–191-S10, 30–191-S16, 30–191-S17), Chom Phra (strains 30–194-S03, 30–194-S04, 30–194-S14), and Sanom (strains 30–198-S22, 30–198-S23, and 30–198-S28). Bacterial manipulations were performed using biosafety level 3 (BSL-3) containment and handled according to institutional and national biosafety guidelines. Bacterial cultures were propagated in Luria-Bertani (LB) broth or on LB agar at 37°C unless otherwise specified. For long-term preservation, stocks were maintained in 20% glycerol at -80°C.

## Comparison of Phosphate-buffered acidic erythritol (ACER) and Ashdown agar for culturing *B. pseudomallei*

The efficiency of different culture media for supporting growth of *B. pseudomallei* was initially evaluated by comparing two agar-based media: Ashdown agar [20] and ACER agar [12]. Bacterial growth was evaluated using ten environmental *B. pseudomallei* strains (30–191-S08, 30–191-S10, 30–191-S16, 30–191-S17, 30–194-S03, 30–194-S04, 30–194-S14, 30–198-S22, 30–198-S23, and 30–198-S28) and the reference clinical strain K96243. The bacterial strains were obtained from frozen stocks, sub-cultured on Columbia agar (Oxoid, UK), and incubated at 37°C for 24 hours. The resulting colonies were re-suspended in PBS, adjusted to an optical density 600 nm ($OD_{600}$) of 0.18 (approximately $1 \times 10^8$ CFU/ml), and diluted to a final concentration of $1 \times 10^3$ CFU/ml. One hundred microliters of each of the suspensions was plated onto ACER agar and Ashdown agar in triplicate and incubated at 37°C for 3 and 7 days. The resulting colonies were enumerated, and their morphology was examined.

## Comparison of *B. pseudomallei* recovered from spiked soil samples in different broth media

The efficiency of ACER, TBSS-C50-based erythritol, and TBSS-C50 media for the recovery of eleven different *B. pseudomallei* strains from spiked soil samples was evaluated as previously described [8,10,12]. Ten environmental strains (30–191-S08, 30–191-S10, 30–191-S16, 30–191-S17, 30–194-S03, 30–194-S04, 30–194-S14, 30–198-S22, 30–198-S23, and 30–198-S28) and the reference clinical strain K96243 were included in this study. Bacterial strains were sub-cultured on Columbia agar (Oxoid, UK) and incubated at 37°C for 24 hours. The resulting colonies were re-suspended in PBS, adjusted an $OD_{600}$ to obtain approximately $1 \times 10^8$ CFU/ml. One ml of each strain in PBS was spiked into 5 g of autoclaved soil, followed by the addition of 9 ml of PBS. The samples were then incubated at room temperature for 24 hours. After incubation, 500 µl of each of the soil supernatants was transferred into 10 ml of ACER, TBSS-C50-based erythritol, or TBSS-C50 enrichment broth, in duplicate, and then incubated under static conditions at 37°C with loosely capped tubes for 2, 5, and 9 days. At each time point, 100 µl of each broth culture was serially diluted (10-fold) in PBS, spread onto Ashdown agar in duplicate, incubated at 37°C for 4 days, following which bacterial colonies were enumerated.

## Detection of *B. pseudomallei* from spiked samples using real time-PCR assays

**DNA extraction.** For real-time PCR assays, genomic DNA was extracted from 200 µl of bacterial pellet obtained from 1 ml of *B. pseudomallei* K96243 culture in LB broth or 250 mg of autoclaved soil spiked with ten environmental *B. pseudomallei* strains using the PowerSoil Pro Kit (QIAGEN) following the manufacturer's instructions. Briefly, the samples were lysed in Solution CD1, vortexed, and centrifuged. The resulting supernatants were treated with Solution CD2 for inhibitor removal, followed by DNA binding with Solution CD3 using an MB Spin Column. Columns were washed with Solutions EA and C5, centrifuged to remove residual wash buffer, and DNA was eluted with 50 µl of Solution C6. The purified DNA samples were stored at -80°C until needed for real-time PCR analysis.

**Real-time PCR assay.** TaqMan real-time PCR assays were evaluated for the detection of *B. pseudomallei* using the *TTS1-orf2* [15] and *BPSS1187* [21] as the target genes. The assays were tested using purified genomic DNA from strain K96243 and the ten environmental isolates. The total reaction volumes were 10 µl and contained 5µl of 1X SensiFAST Probe No-ROX kit, 400 nM each of forward and reverse primers, 100 nM of the probe, and 1 µl of total genomic DNA. The

amplification conditions for *BPSS1187* included an initial denaturation step at 95°C for 5 minutes, followed by 45 cycles of 95°C for 15 seconds and 58°C for 30 seconds. For *TTS1-orf2*, the conditions consisted of an initial denaturation at 95°C for 8 minutes, followed by 45 cycles of 95°C for 15 seconds and 59°C for 15 seconds. All reactions were performed in a CFX96 Touch Real-time PCR system and analyzed using Bio-Rad CFX Maestro software, version 4.1.2433.1219 (Bio-Rad).

Data analysis was performed using serial 10-fold dilutions of genomic DNA, corresponding to a range of $10^8$ to 1 CFU/ml. Standard curves plotted threshold cycle ($C_t$) values against the log of bacterial concentration (CFU/ml). Linear regression analysis was then applied to the data to assess the assay efficiency and dynamic range. PCR efficiency was calculated from the slope of the standard curve using the equation of $E = 10^{-1/slope}-1$ [22].

**Determination of the limit of detection of real-time PCR assays.** The limit of detection (LOD) of the real-time PCR assays was evaluated using specific primers targeting *TTS1-orf2* and *BPSS1187* in a TaqMan real-time PCR format [18]. The assays were performed using genomic DNA from *B. pseudomallei* K96243 in PBS and ten environmental strains spiked into soil samples. DNA was extracted as described above and concentrations were quantified using a NanoDrop Lite spectrophotometer (Thermo Fisher Scientific, Waltham, MA, USA). Genomic DNA corresponding to CFU/ml ranging from $10^8$ to 1 CFU/ml was analysed by real-time PCR. The LOD was defined as the lowest concentration at which amplification was consistently observed across three replicates and was expressed as CFU/ml.

## Enzyme linked immunosorbent assay for detection of Hcp1-specific antibodies in plasma collected from healthy individuals

Hcp1-specific IgG antibodies were measured in 398 plasma samples collected from healthy individuals in a previous study conducted in Amnat Charoen province using a single point (1:2000 dilution) enzyme linked immunosorbent assay (ELISA), as previously described [23]. Two cut-off values were applied to define seropositivity. The first cut-off value calculated as mean $OD_{450nm}$ plus 1.5 standard deviation (OD = 0.418) was used to identify individuals with probable exposure to *B. pseudomallei*. The second cut-off (OD 1.165), as previously described [24], was used to identify individuals with probable melioidosis.

## Soil sampling

A total of 238 soil samples were collected from three districts in Amnat Charoen Province, Northeast Thailand in November 2024. Sampling locations were selected based on seroprevalence data from healthy populations, targeting households of healthy individuals who were strongly seropositive to *B. pseudomallei* (N = 6), as determined by Hcp1-specific ELISA, while households of seronegative individuals (N = 6) were randomly selected for soil sampling. The sampling sites included Mueang Amnat Charoen district (N = 109), Pathum Ratchawongsa district (N = 60), and Chanuman district (N = 69).

Soil sampling was performed as previously described [25]. Briefly, the soil samples were collected from a depth of 30 cm, with sampling points spaced 5–10 meters apart. A steel auger was used for soil collection and was cleaned with bottled water and disinfected with 70% ethanol between each sample to prevent cross-contamination. For each sampling point, 200 g of soil was placed in plastic zip lock bags, stored at ambient temperature, protected from direct sunlight, and transported to laboratory (Faculty of Tropical Medicine, Mahidol University, Bangkok) within two days.

## Culture of *B. pseudomallei* from soil samples

For isolation of *B. pseudomallei* from soil samples, two culture approaches were performed using established protocols: 1) direct plating culture on Ashdown agar [25–27] and 2) culture in ACER enrichment broth [8,10,12,27]. For direct plating on agar, 100 g of soil from each sample was suspended in 100 ml of sterile water, mixed and incubated at room temperature overnight. Subsequently, two aliquots of 10 µl and 100 µl of the soil supernatants were plated onto Ashdown agar in

duplicate and incubated at 42°C for 3 days. Bacterial colonies were then counted. Suspected *B. pseudomallei* colonies were confirmed using latex agglutination [28] and real-time PCR targeting *TTS1-orf2* gene [15].

For culture in enrichment broth [12], five grams of soil from each sample was homogenized in 10 ml of PBS, and the suspensions were left to settle overnight at room temperature. A 500 µl aliquot of supernatant was then inoculated into 10 ml of ACER broth and incubated statically with a loose cap at 40°C for 5 and 9 days [12]. Following incubation, 100 µl of broth culture was 10-fold serial diluted, 100 µl of each dilution was plated onto Ashdown agar, and the plates were incubated at 40°C for 4 days. Bacterial colonies were counted and confirmed to be *B. pseudomallei* [15,28] as described above.

### Real-time PCR of soil samples positive for *B. pseudomallei* by culture-based methods

A total of 20 soil samples that tested positive for *B. pseudomallei* by culture-based methods were extracted as described above. All samples were subsequently analyzed using real-time PCR assays targeting *BPSS1187* and *TTS1-orf2* genes.

### Statistical analysis

Statistical analyses were performed using GraphPad Prism version 9.5.1 (GraphPad Software, San Diego, CA, USA). Comparisons of bacterial colony counts between different media were conducted using the Wilcoxon signed-rank test. The efficiencies of real-time PCR assays targeting the *BPSS1187* and *TTS1-orf2* genes were compared using a paired t-test. The association between the prevalence of *B. pseudomallei* and seroprevalence was assessed using the Fisher's exact test. The Chi-square test was used to analyze the demographic distribution of individuals with Hcp1-specific antibody positivity.

## Results

### Comparison of Ashdown and ACER agar for culture of *B. pseudomallei*

The recovery of *B. pseudomallei* from PBS was assessed by culturing ten environmental strains and the clinical strain K96243 on Ashdown and ACER agar in three independent experiments (Fig 1). At day 3 post-inoculation, *B. pseudomallei* colonies were detectable on both media. For all experiments, the median CFU per plate on ACER agar was not significantly different from the median CFU per plate on Ashdown agar (Fig 1A and S1 Table). Colonies on Ashdown agar exhibited type I morphotypes [29], characterized by a rough central surface with irregular margins and a pale purple appearance, with a diameter ranging from 1 to 2 mm. In contrast, colonies on ACER agar appeared smooth, creamy, white, and opaque, with a diameter of approximately 0.1 to 0.5 mm (Figs 2 and S1).

At day 7 post-inoculation, the median CFU per plate on ACER agar was not significantly different from that of Ashdown agar. The number of colonies and morphologies were also the same as on day 3 (Fig 1B and S1 Table). In contrast, the average colony size (diameter) on Ashdown agar had increased to 5 mm but was only 1 mm on ACER agar Figs 2 and S1). Since the recovery of all strains of *B. pseudomallei* on Ashdown agar was similar and there was a more distinct colony morphology compared to ACER agar, Ashdown was selected for further experiments and used for environmental surveillance of *B. pseudomallei.*

### Comparison of three different enrichment media for the recovery of *B. pseudomallei* from spiked soil samples

The recovery of ten environmental *B. pseudomallei* strains and the reference clinical strain K96243 from spiked soil samples was assessed using TBSS-C50, TBSS-C50-based erythritol, and ACER broth following 2, 5, and 9 days of incubation. Colony counts were determined by plating serial dilutions on Ashdown agar (Fig 3 and S2 Table). *B. pseudomallei* was recovered from all three types of broth at all time points, but ACER broth demonstrated a higher bacterial yield than TBSS-C50-based erythritol and TBSS-C50 broths. The median colony counts for ACER broth were significantly higher than those for TBSS-C50 broth (*p* = 0.001), while TBSS-C50-based erythritol broth showed the lowest recovery for all time points.

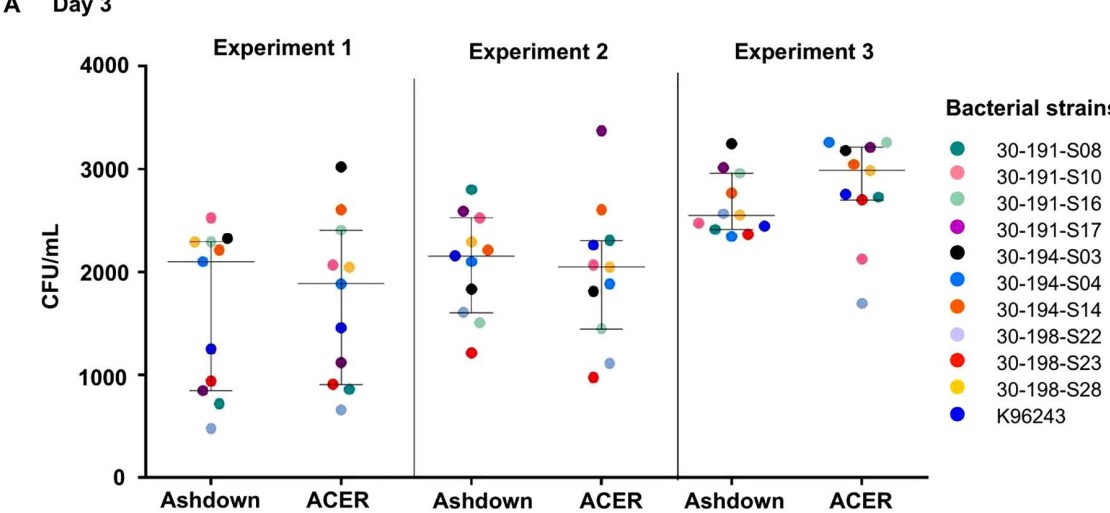

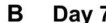

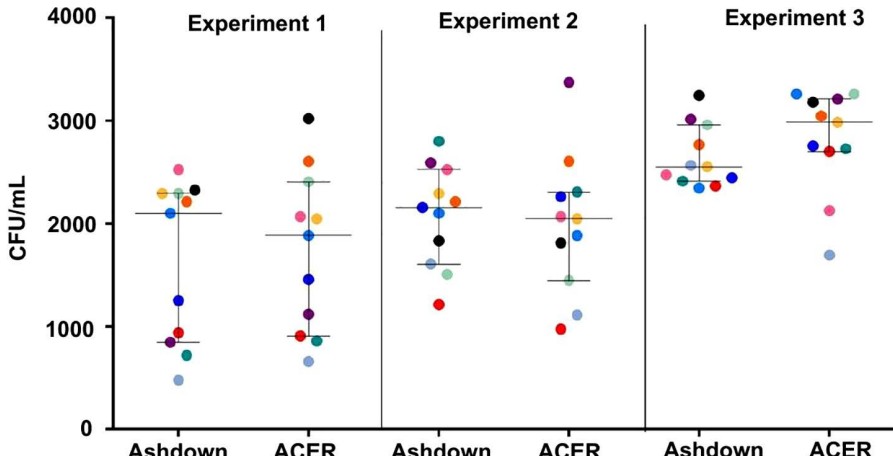

**Fig 1. Colony counts of eleven *B. pseudomallei* strains on Ashdown agar and ACER agar.** The number of colonies of *B. pseudomallei* were counted at (A) 3 days and (B) 7 days of incubation at 37 °C. The experiments were performed in triplicate in three independent assays.

## Comparison of real-time PCR methods for detecting *B. pseudomallei* in PBS

We evaluated the performance of real-time PCR assays targeting *BPSS1187* and *TTS1-orf2* for detecting *B. pseudomallei* strain K96243 in PBS. Both targets exhibited strong linear correlations between log CFU/ml and $C_t$ values. For *BPSS1187*, the average regression equation was $y = -3.414x + 43.24$ with an $R^2$ of 0.9856 and a LOD of $10^2$ CFU/ml. For *TTS1-orf2*, the average regression equation was $y = -5.447x + 56.39$ with an $R^2$ of 0.9734 and a LOD of $10^4$ CFU/ml (Fig 4 and S3 and S4 Tables). A paired-t test was used to compare PCR efficiencies across three replicates for *BPSS1187*-PCR (87.6%, 106.1%, and 96.9% efficiency) and *TTS1-orf2*-PCR (51.0%, 54.7%, and 51.5% efficiency). The analysis revealed significant differences in efficiency between the two assays, with a median efficiency of 96.9% for *BPSS1187* and 51.5% for *TTS1-orf2*, representing a difference of 45.4% ($p = 0.009$).

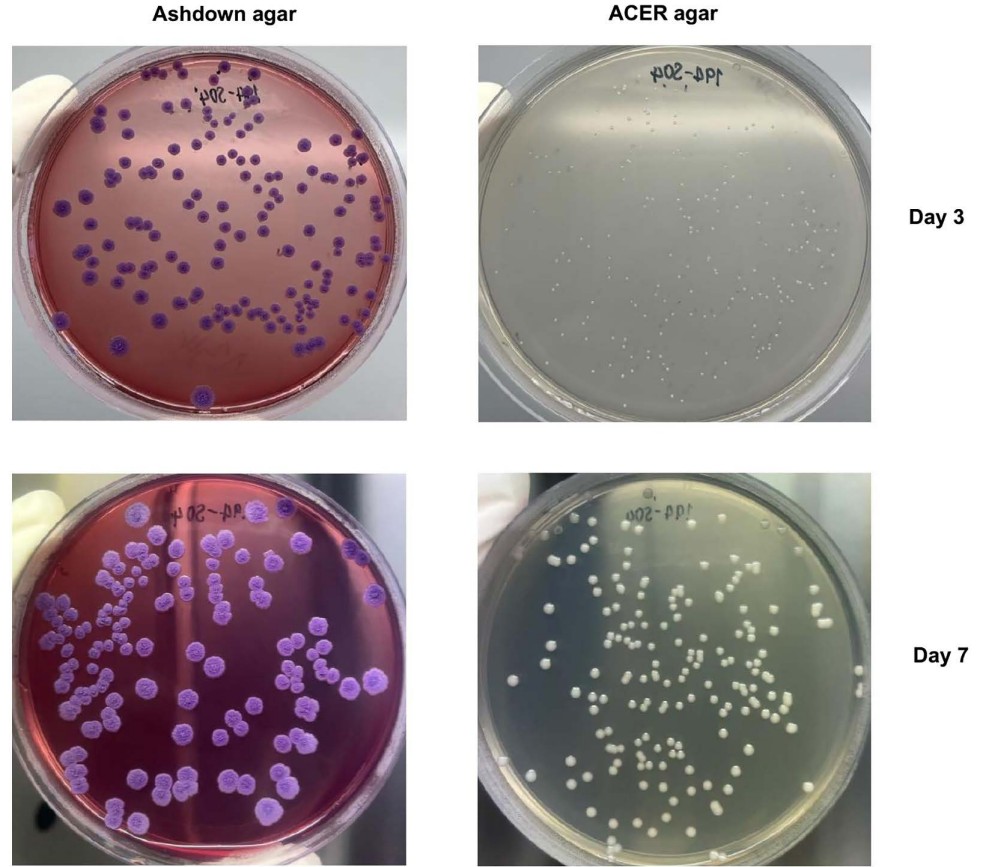

**Fig 2. Colony morphology of _B. pseudomallei_ on Ashdown agar and ACER agar.** The colony morphology of _B. pseudomallei_ 30-194-S04 was examined at 3 days and 7 days of incubation on Ashdown and ACER agar respectively.

## Comparison of real-time PCR methods for detecting _B. pseudomallei_ in spiked soil samples

Next, we evaluated the performance of real-time PCR assays targeting _BPSS1187_ and _TTS1-orf2_ for detecting ten environmental _B. pseudomallei_ strains spiked into soil samples. Calibration curves were generated for each strain and demonstrated high consistency across assays ([Fig 5]). For _BPSS1187_, the slopes ranged from –3.409 to –4.065, with $R^2$ values between 0.9469 and 0.9977, and PCR efficiencies ranging from approximately 77% to 96%. For _TTS1-orf2_, the slopes ranged from –2.847 to –4.203, $R^2$ values from 0.6819 to 0.999, and PCR efficiencies from approximately 77% to over 111%. To statistically compare the PCR efficiencies of the two targets amongst the 10 environmental strains, a paired t-test was performed. The analysis revealed no statistically significant differences in the efficiencies between the two targets with a median efficiency of 84.6% for _BPSS1187_ and 87.4% _TTS1-orf2,_ which was a difference of only 2.8% ($p = 0.188$) ([S4] and [S5 Tables])

## Hcp1-specific antibody responses in healthy individuals in Amnat Charoen province

To investigate the environmental distribution of _B. pseudomallei_ in relation to potential human exposures, we first determined the Hcp1-specific IgG responses in 398 plasma samples from healthy individuals living in seven different districts

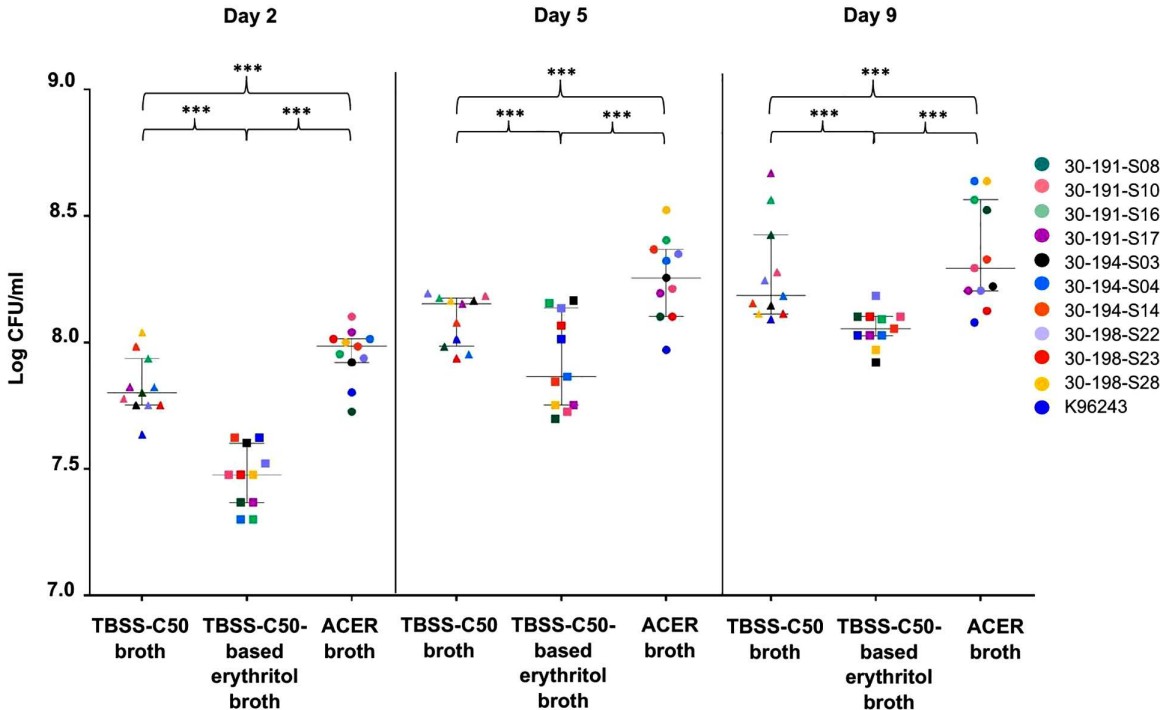

**Fig 3. Bacterial counts of eleven *B. pseudomallei* strains on TBSS-C50, TBSS-C50-based erythritol medium, and ACER broth.** Eleven *B. pseudomallei* strains were grown statically at 37°C in TBSS-C50, TBSS-C50-based erythritol medium, and ACER broth for 2, 5, and 9 days. Each point represents one of the *B. pseudomallei* strains (***$p < 0.001$; Wilcoxon Signed-Rank Test).

of Amnat Charoen province. Seropositivity was defined using two cut-off values $OD_{450} = 0.418$ (probable exposure to *B. pseudomallei*) and $OD_{450} = 1.165$ (probable melioidosis), established in a previous study [24]. Among the 398 samples, 34 individuals (8.5%) were identified as seropositive ($OD \geq 0.418$). Of these, 28 individuals were classified as having probable exposure to *B. pseudomallei*, while 6 individuals were classified as having probable melioidosis ($OD \geq 1.165$) (Fig 6 and Tables 1 and S6).

Seropositive individuals with probable melioidosis were identified in Chanuman (2/44, 4.5%), Mueang Amnat Charoen (3/136, 2.2%), and Pathum Ratchawongsa (1/48, 2.1%). While seropositive individuals with probable exposure to *B. pseudomallei* were more widely distributed and included Phana (4/31, 12.9%), Pathum Ratchawongsa (5/48, 10.4%), Senangkhanikhom (5/48, 10.4%), Lue Amnat (3/38, 7.9%), Hua Taphan (3/53, 5.6%), Chanuman (2/44, 4.5%), and Mueang Amnat Charoen (6/136, 4.4%). All 34 seropositive individuals were distributed across age groups, with a seropositivity rate of 7.5% among individuals aged 13–17 years (N=53), 7.5% among those aged 18–59 years (N=253), and 12.0% among those aged 60 years or older (N=92). However, no statistically significant association was found between age group and seroprevalence (*P*=0.410). By gender, seropositivity was 11.5% in males and 7.8% in females. No statistically significant association was observed between gender group and seroprevalence (*P*=0.291). Differences by occupation were also observed, with seropositivity rates of 8.9% among students, 9.9% among farmers, and 6.3% among unemployed individuals. No statistically significant association was observed between occupation group and seroprevalence (*P*=0.522) (Table 2).

## Soil sampling in areas surrounding the households of healthy individuals

Hcp1-ELISA results were used to guide the selection of soil sampling locations around the houses of individuals that were strongly positive (N=6) and negative (N=6) for Hcp1-specific antibodies. To investigate the prevalence of *B. pseudomallei*

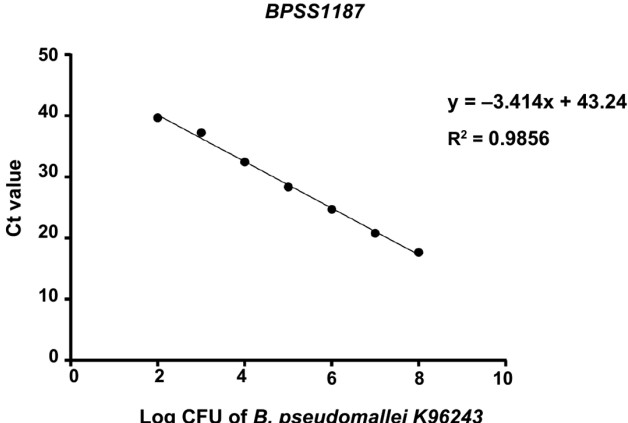

**BPSS1187**

$$y = -3.414x + 43.24$$
$$R^2 = 0.9856$$

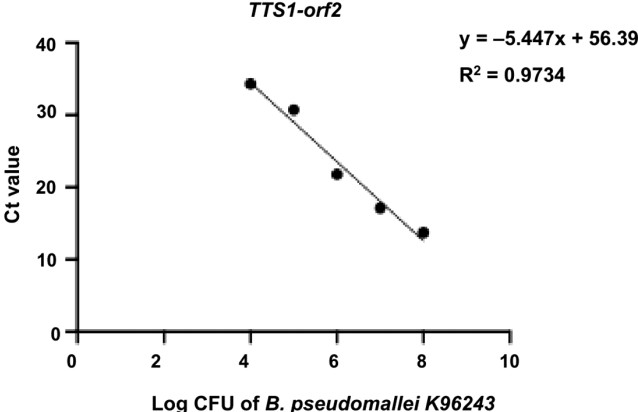

**TTS1-orf2**

$$y = -5.447x + 56.39$$
$$R^2 = 0.9734$$

**Fig 4. Analytical sensitivity of real-time PCR assays for *B. pseudomallei* K96243.** Calibration plots of bacterial concentration (log CFU/ml) versus cycle threshold ($C_t$) values for the *BPSS1187* and *TTS1-orf2* targets across nine serial dilutions ($10^8$ to $10^0$ CFU/ml).

in soil samples, we employed two approaches: (I) direct plating on Ashdown agar and (II) enrichment culture in ACER broth prior to plating on Ashdown agar. The percentage of *B. pseudomallei*-positive samples varied across methods and districts. For direct plating on Ashdown agar, the highest detection rate was in Pathum Ratchawongsa, where 4/60 samples (6.7%) tested positive, followed by Mueang Amnat Charoen where 3/109 samples (2.8%) tested positive, and Chanuman where 1/69 samples (1.5%) tested positive. For ACER enrichment culture, positive results on day 5 were detected in 5/60 samples (8.3%) from Pathum Ratchawongsa, 6/109 samples (5.5%) from Mueang Amnat Charoen, and 1/69 sample (1.5%) from Chanuman. By day 9, the number of positive samples increased in Mueang Amnat Charoen (8 of 109; 7.3%), remained unchanged in Pathum Ratchawongsa and Chanuman (Fig 7 and Tables 3 and S7).

Of these 20 positive samples, 5 samples were positive in all culture methods. Two samples were positive only by direct plating on Ashdown agar, 4 samples were positive only after enrichment culture in ACER for 5 days, and 5 samples were positive only after enrichment culture in ACER for 9 days (Fig 8 and S8 Table).

A total of 20 soil samples confirmed to be culture positive for *B. pseudomallei* were further analyzed using real-time PCR assays targeting the *BPSS1187* and *TTS1-orf2* genes. The results showed that 2 out of 11 samples from Mueang Amnat Charoen district, and none of the samples from Pathum Ratchawongsa (0/8) and Chanuman (0/1) districts, tested positive for both genes (Table 3).

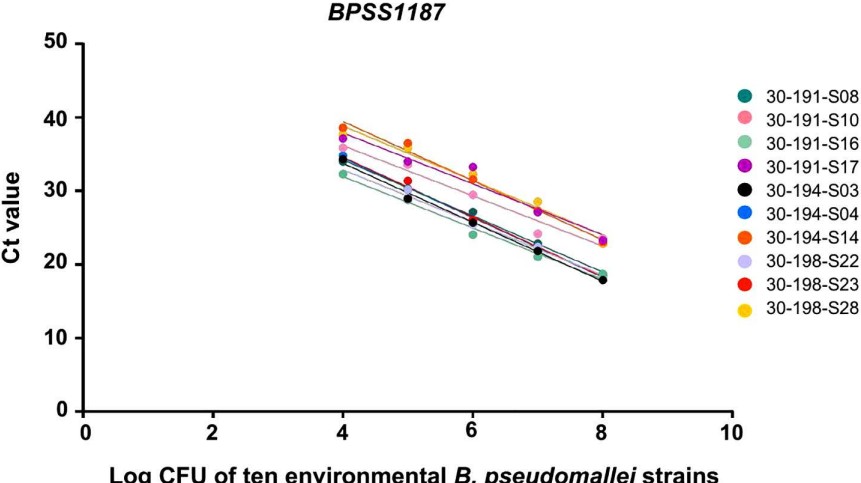

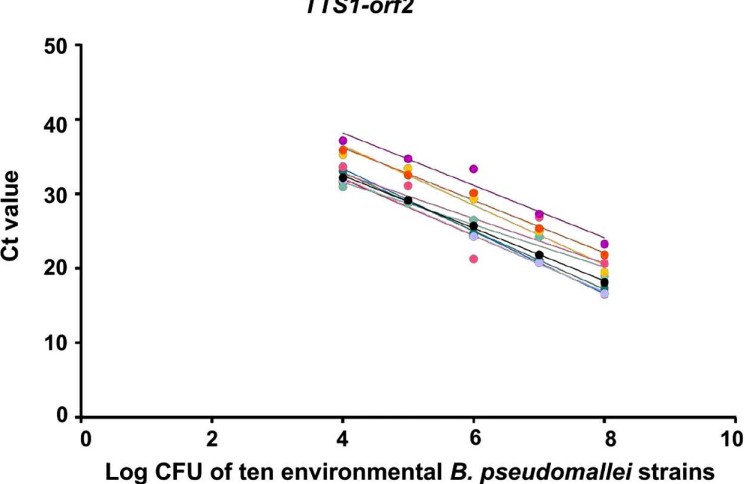

**Fig 5. Analytical sensitivity of real-time PCR assays for 10 environmental *B. pseudomallei* strains.** Calibration plots of bacterial concentration (log CFU/ml) versus cycle threshold (Ct) values for *BPSS1187* and *TTS1-orf2* across nine serial dilutions ($10^8$ to $10^0$ CFU/ml).

The base map was generated using the open-source R packages rgdal and rworldmap, with administrative boundary shapefiles obtained from UN-OCHA (https://data.humdata.org/dataset/cod-ab-tha). The shapefile is freely available for scientific, educational, and commercial use under the Creative Commons Attribution 4.0 International License (CC BY 4.0) (https://creativecommons.org/licenses/by/4.0/).

### The association between the presence of *B. pseudomallei* in the soil and seroprevalence

To evaluate the association between environmental detection of *B. pseudomallei* and seroprevalence, Fisher's exact test was performed using data from 12 household soil samples, consisting of individuals that were strongly positive (N=6) and negative (N=6) for Hcp1-specific antibodies, as summarized in **Table 4**. Based on combined culture-based methods, *B. pseudomallei* was detected in three households associated with strongly seropositive individuals and in two households associated with seronegative individuals. The analysis revealed no significant association across the three districts ($P$=0.165), indicating no statistically significant association between the presence of *B. pseudomallei* in the household environment and seroprevalence.

**(A)**

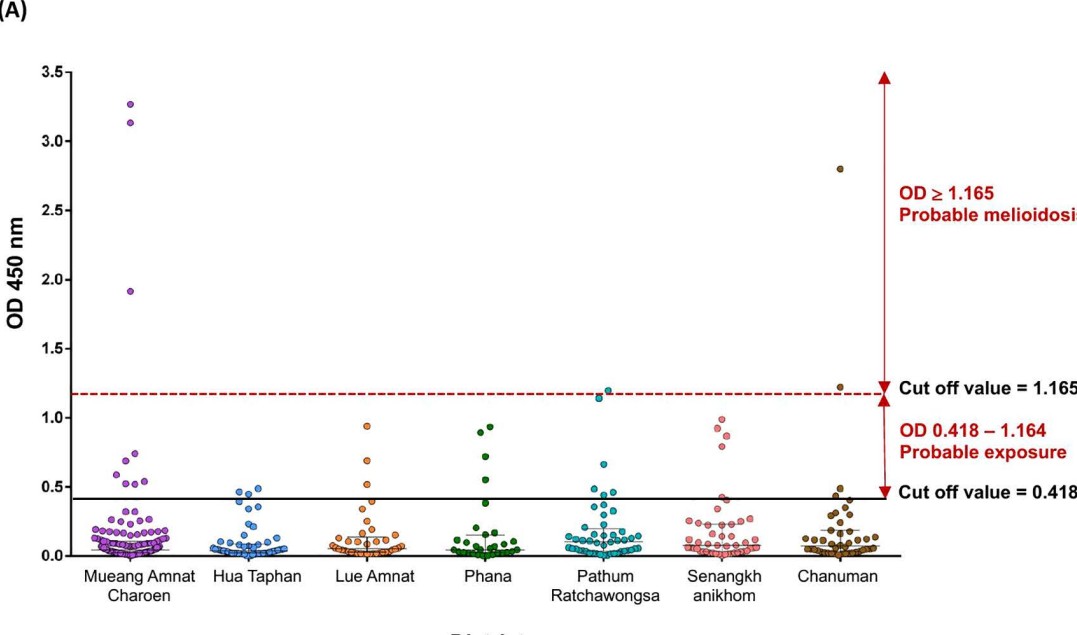

**(B)**

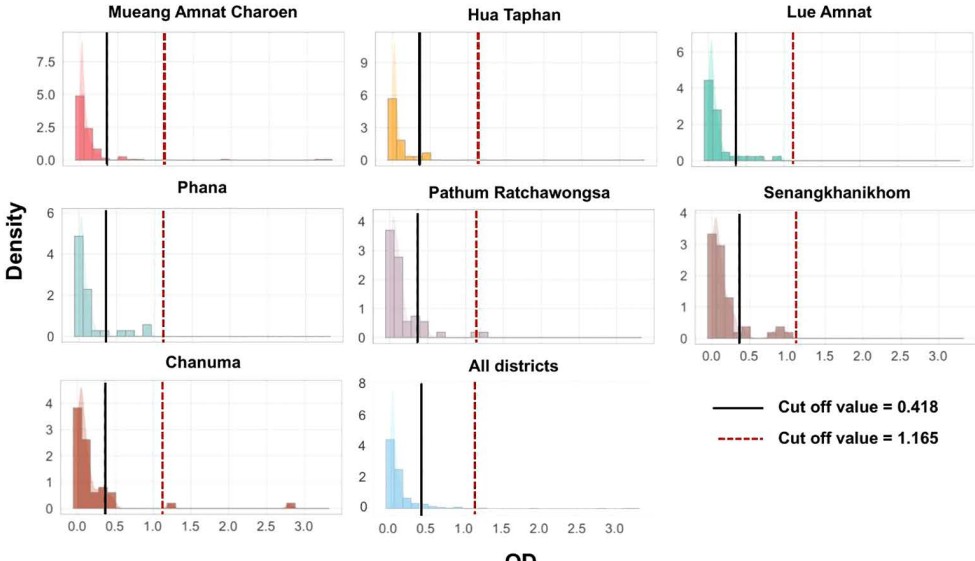

**Fig 6. Distribution of antibody responses among 398 healthy individuals across seven districts of Amnat Charoen Province. (A)** Scatter plot of antibody responses across seven districts of Amnat Charoen province. **(B)** Histograms showing the distribution of antibody responses within each district and across all districts combined. The solid black line represents the cut off value (0.418), while the dashed line marks the cut off value (1.165).

Neglected Tropical Diseases

**Table 1. Detection of Hcp1-specific IgG among healthy individuals from seven districts in Amnat Charoen Province, Thailand.**

| District | Total serum samples | Total positive individuals | Individuals with seropositive (%) | |
|---|---|---|---|---|
| | | | Probable exposure | Probable melioidosis |
| Mueang Amnat Charoen | 136 | 9 (6.6%) | 6 (4.4%) | 3 (2.2%) |
| Hua Taphan | 53 | 3 (5.6%) | 3 (5.6%) | 0 |
| Lue Amnat | 38 | 3 (7.9%) | 3 (7.9%) | 0 |
| Phana | 31 | 4 (12.9%) | 4 (12.9%) | 0 |
| Pathum Ratchawongsa | 48 | 6 (12.5%) | 5 (10.4%) | 1 (2.1%) |
| Senangkhanikhom | 48 | 5 (10.4%) | 5 (10.4%) | 0 |
| Chanuman | 44 | 4 (9.0%) | 2 (4.5%) | 2 (4.5%) |
| Total | 398 | 34 (8.5%) | 28 (7.0%) | 6 (1.5%) |

A total of 398 plasma samples were obtained from healthy individuals across seven districts. The presence of Hcp1-specific IgG was assessed, the number and percentage of positive samples were determined using two cut-off values: OD values ≥ 0.418, defined as probable exposure to *B. pseudomallei*, and OD values ≥ 1.165, defined as probable melioidosis.

**Table 2. Demographic distribution of Hcp1-specific antibody positive individuals.**

| Demographic data | Total individuals (N = 398) | Hcp1-specific IgG antibody (OD450 ≥ 0.418) | | P-value |
|---|---|---|---|---|
| | | Positive samples (%) (N = 34) | Negative samples (%) (N = 364) | |
| ***Age groups (year)*** | | | | 0.410 |
| 13-17 | 53 | 4 (7.5%) | 49 (92.5%) | |
| 18-59 | 253 | 19 (7.5%) | 234 (92.5%) | |
| ≥ 60 | 92 | 11 (12%) | 81 (89%) | |
| ***Gender*** | | | | 0.291 |
| Male | 78 | 9 (11.5%) | 69 (88.5%) | |
| Female | 320 | 25 (7.8%) | 295 (92.2%) | |
| ***Occupation*** | | | | 0.523 |
| Student | 45 | 4 (8.9%) | 41 (91.1%) | |
| General contract occupation | 20 | 0 (0%) | 20 (100%) | |
| Farmer | 291 | 29 (10%) | 262 (90%) | |
| Government officer | 10 | 0 (0%) | 10 (100%) | |
| Unemployed | 16 | 1 (6.3%) | 15 (93.7%) | |
| Merchant | 2 | 0 (0%) | 2 (100%) | |
| No data | 14 | 0 (0%) | 14 (100%) | |

The Chi-square test was used to assess the association between Hcp1-specific IgG seroprevalence and demographic variables. The table presents the number and percentage of Hcp1-specific IgG-positive and -negative samples across different demographic groups, categorized by age group, gender, and occupation.

## Discussion

Our evaluation of *B. pseudomallei* detection methods provides critical insights for optimizing environmental surveillance strategies in an endemic area of Northeast Thailand, through a comprehensive comparative analysis of culture- and molecular-based techniques on soil samples and the association between prevalence of *B. pseudomallei* and seroprevalence. Comparison of the growth of *B. pseudomallei* on Ashdown and ACER agar showed that Ashdown agar performed better than ACER agar in terms of growth and provided a more distinct colony morphology of *B. pseudomallei*. These findings are consistent with previous reports, which emphasized using Ashdown agar for presumptive identification, based on

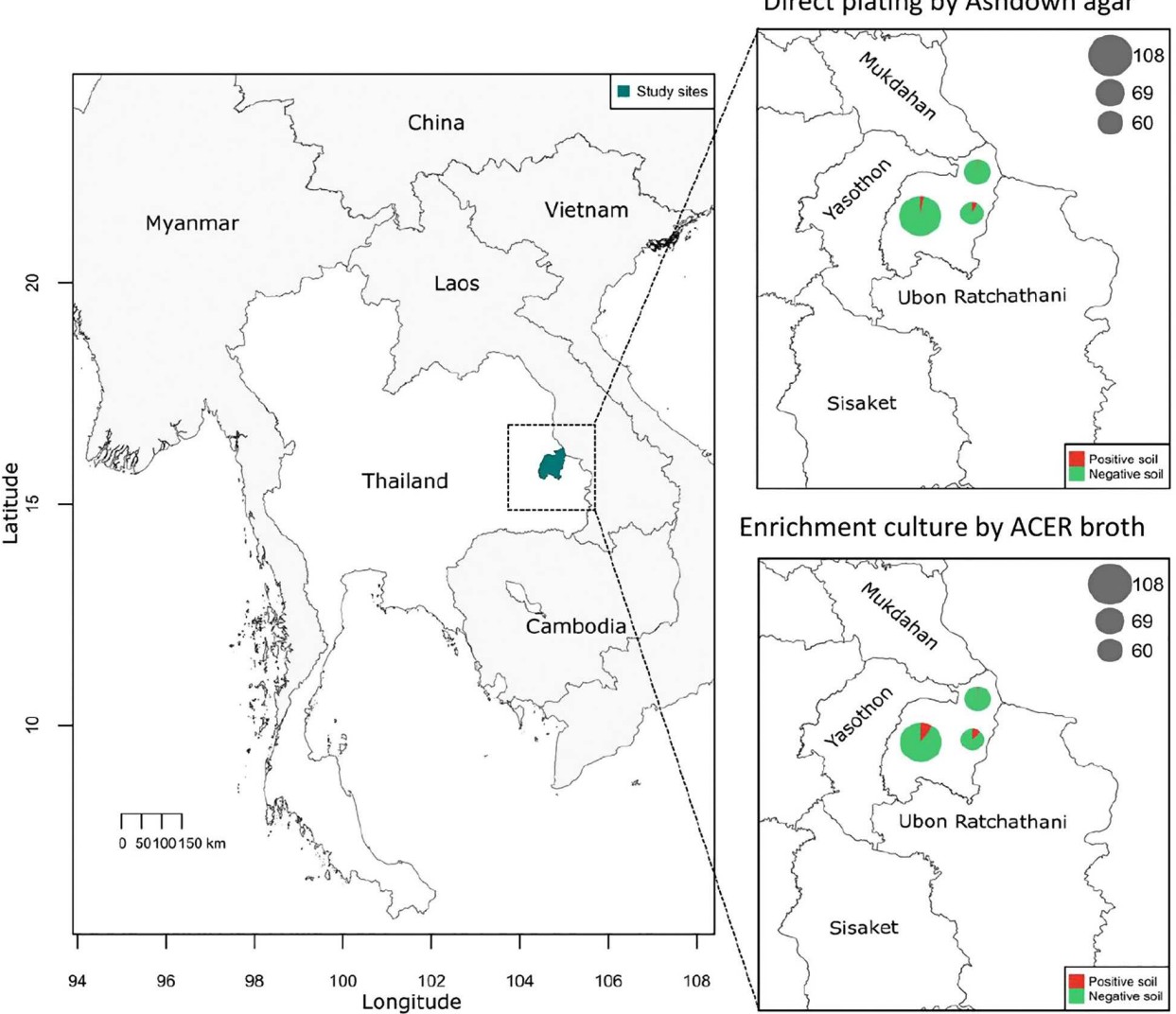

**Fig 7. Geographical distribution of *B. pseudomallei* detected from soil samples in Amnat Charoen Province, Thailand.** The main map highlights sampling sites within the province. Enlarged panels show culture results on Ashdown agar (top right) and ACER broth (bottom right) across three districts. Each pie chart represents the proportion of positive (red) and negative (green) samples, with size proportional to the number of samples collected (Mueang Amnat, n = 108; Pathum Ratchawongsa, n = 60; Chanuman, n = 69).

the distinct wrinkled and purple colony appearance characteristic of *B. pseudomallei* [29]. This particular colony appearance is attributed to components of Ashdown agar, including crystal violet, which contributes to the purple coloration of *B. pseudomallei* colonies, and neutral red, which enhances their distinctive wrinkled appearance [20]. Although ACER agar supported comparable bacterial counts, differentiation was challenging due to the less distinct colony form, necessitating confirmation tests.

Further comparison of the efficacy of TBSS-C50 broth, TBSS-C50-based erythritol broth, and ACER broth in culturing *B. pseudomallei* demonstrated that ACER broth provided superior enrichment performance, yielding a higher recovery rate of *B. pseudomallei* compared to both TBSS-C50-based erythritol broth and TBSS-C50 broth. This is consistent with

**Table 3. Culture- and real-time PCR-based detection of *B. pseudomallei* in soil samples from three districts of Amnat Charoen Province, Thailand.**

| Districts | No. of houses | No. of soil samples for culture | No. of samples positive *B. pseudomallei* by culture | | | No. of culture positive samples for real-time PCR assay | No. of samples positive by real-time PCR | |
|---|---|---|---|---|---|---|---|---|
| | | | Direct plating on Ashdown agar | ACER enrichment broth | | | | |
| | | | | Day 5 | Day 9 | | *BPSS1187* | *TTS1-orf2* |
| Mueang Amnat Charoen | 5 | 109 | 3 (2.8%) | 6 (5.5%) | 8 (7.4%) | 11 | 2 (18.2%) | 2 (18.2%) |
| Pathum Ratchawongsa | 3 | 60 | 4 (6.7%) | 5 (8.3%) | 5 (8.3%) | 8 | 0 | 0 |
| Chanuman | 4 | 69 | 1 (1.5%) | 1 (1.5%) | 0 | 1 | 0 | 0 |
| Total | 12 | 238 | 8 (3.4%) | 12 (5.1%) | 13 (5.5%) | 20 | 2 (10%) | 2 (10%) |

The table presents the number and percentage of *B. pseudomallei*-positive soil samples detected by culture-based and real-time PCR methods. Soil samples were collected from Mueang Amnat Charoen, Pathum Ratchawongsa, and Chanuman districts.

**Fig 8. Detection of *B. pseudomallei* in soil samples by culture and PCR methods.** Euler diagram illustrating the overlap of results among culture-based and PCR-based detection methods for 20 *B. pseudomallei*-positive soil samples collected in Amnat Charoen Province, Thailand.

**Table 4. Seroprevalence and culture-based detection of *B. pseudomallei* in household soil samples across three districts in Amnat Charoen Province, Thailand.**

| Districts | Seropositive | Seronegative | *P*-value |
|---|---|---|---|
| Mueang Amnat Charoen | 5/60 (8.3%) | 6/48 (12.5%) | |
| Pathum Ratchawongsa | 8/45 (17.8%) | 0/15 (0.0) | |
| Chanuman | 1/24 (4.2%) | 0/45 (0.0) | |
| Total | 14/129 (10.9%) | 6/108 (5.5%) | 0.165 |

The association between seroprevalence and culture-based detection of *B. pseudomallei* for household soil samples collected from three districts: Mueang Amnat Charoen, Pathum Ratchawongsa, and Chanuman. Individuals were classified as seropositive or seronegative based on Hcp1-specific IgG levels. A cut-off $OD_{450nm}$ value greater than 1.165 was used.

previous studies reporting that ACER broth enhances the recovery of *B. pseudomallei*, likely due to its acidic condition (pH 6.3), which shortens the lag phase of bacterial growth [12]. While *B. pseudomallei* grows optimally at neutral pH, acidic conditions (approximately pH 5.0–5.5) have been shown to enhance recovery from environmental samples by suppressing competing organisms and reducing the lag phase [20,30]. Additionally, erythritol, used as a carbon source in enrichment media, has been demonstrated to promote *B. pseudomallei* proliferation and improve recovery rates from environmental samples [12,31].

Our study evaluated the efficiency of real-time PCR assays targeting *BPSS1187* and *TTS1-orf2* for the detection of *B. pseudomallei*. The PCR assay targeting *BPSS1187* demonstrated significantly higher efficiency than the *TTS1-orf2* assay when performed on PBS samples spiked with *B. pseudomallei* strain K96243. This result is consistent with a previous study that reported high sensitivity and specificity for *B. pseudomallei* detection in plasma samples using real-time PCR targeting *BPSS1187* [18]. However, no statistically significant difference in PCR efficiency was observed between the two targets across multiple environmental strains. These findings suggest that both assays exhibited comparable performance for the detection of diverse *B. pseudomallei* strains in environmental samples.

We also investigated Hcp1-specific antibody responses to *B. pseudomallei* among individuals in Amnat Charoen province. Our study revealed an overall seropositivity rate of 8.5% among seropositive individuals that included individuals with probable exposure to *B. pseudomallei* (7.0%) and individuals with probable melioidosis (1.5%). This finding contrasts with a previous study conducted in Ubon Ratchathani, which reported a high seroprevalence rate of 38% among healthy individuals, based on indirect hemagglutination assay (IHA) at a cutoff titer of 1:80 [32]. However, the IHA has been shown in several studies to have low sensitivity and specificity [33,34], which may account for its higher background antibodies to non-specific antigens. In contrast, the Hcp1-ELISA has demonstrated greater sensitivity and specificity for detecting exposure to *B. pseudomallei* [24,34]. In terms of age distribution, our study found no significant differences in seropositivity among age groups. This contrasts with previous findings from northeast Thailand, where high antibody titers were detected in approximately 20% of 2,214 children aged <15 years in Udon Thani and 11% of children in Ubon Ratchathani, as determined using the IHA [35], potentially reflecting increased environmental exposure through outdoor school activities or contact with contaminated soil and water. Regarding gender, our findings also showed no significant difference in seropositivity. This contrasts with prior studies that have reported higher seroprevalence among males [36], attributed to differences in testing methods (IHA versus Hcp1-ELISA), behavioral factors such as outdoor occupations, greater exposure to soil and water, and alcohol consumption. Additionally, our results demonstrated no significant differences in seropositivity across occupational groups. This contrasts with previous studies that have reported significantly higher seroprevalence among farmers compared to non-farmers [32].

In addition, we collected soil from areas surrounding the households of individuals that were strongly positive and negative based on Hcp1-specific IgG levels to determine the prevalence of *B. pseudomallei* in the environment. Our

study showed that enrichment culture in ACER broth generally outperformed direct plating which is consistent with previous studies demonstrating improved sensitivity of enrichment-based methods for isolation of *B. pseudomallei* in the environment [10,27,37]. Among the surveyed districts, Pathum Ratchawongsa exhibited the highest detection rate using both culture-based approaches. Notably, prolonged incubation in ACER broth (up to nine days) increased the number of positive samples in Mueang Amnat Charoen, highlighting the importance of extended enrichment, especially in samples with low bacterial load [12]. In contrast, Chanuman district showed the lowest detection rate across both methods. These findings suggest that the environmental prevalence of *B. pseudomallei* may be influenced by soil type, consistent with previous reports indicating that silt soils harbor a lower proportion of the bacterium compared to sandy or clay soils [38]. This aligns with our observation in Chanuman, where silt-dominant soils were predominant and relatively compact. In contrast, soil samples from Mueang Amnat Charoen and Pathum Ratchawongsa districts were primarily collected from agricultural areas such as rice paddies, which provide moist and nutrient-depleted conditions favorable for *B. pseudomallei* survival and proliferation [38,39]. Furthermore, seasonal variations influence the prevalence of *B. pseudomallei*. Previous studies have reported that the monsoon season is associated with the highest rates of *B. pseudomallei* detection compared to other seasons [40–43]. Since our study was conducted in November, during the dry season in Thailand, this timing may explain the relatively low detection rates observed, with *B. pseudomallei* identified in only 3.36% of samples by direct plating and 5.04%–5.46% using enrichment methods. Moreover, human activities might also play a role, with previous reports identifying agricultural work, particularly rice farming and gardening, as major risk factors due to the increased disturbance of soil environment [44–46].

We further tested the culture-positive soil samples using real-time PCR targeting the *TTS1-orf2* and *BPSS1187* genes. Of the 20 culture-positive soil samples, *B. pseudomallei* DNA was detected in two samples. This low detection rate may be attributed to the presence of PCR inhibitors commonly found in soil extracts, as well as low DNA yield, since PCR was performed directly on soil without prior enrichment or the use of soil pellet for DNA extraction. In contrast, previous studies have successfully detected *B. pseudomallei* DNA from soil samples by conducting PCR after enrichment using TBSS-C50 based erythritol broth, or ACER broth [10,12] or by extracting DNA from direct soil samples [47,48]. This discrepancy may reflect differences in DNA extraction efficiency, the effectiveness of inhibitor removal, soil composition, and the smaller quantities of soil used for DNA extraction, which may reduce the likelihood of detecting *B. pseudomallei* given its heterogeneous distribution in soil.

Understanding the association between the environmental prevalence of *B. pseudomallei* and human seroprevalence is crucial for informing surveillance strategies in endemic areas. In this study, we investigated whether the presence of *B. pseudomallei* in household soil samples was associated with the individual seroprevalence. However, the results revealed no statistically significant association between environmental detection of *B. pseudomallei* and seroprevalence, and several factors may contribute to the absence of a strong correlation. Seroprevalence reflects cumulative or past exposure, whereas culture-based methods detect the current and often localized presence of *B. pseudomallei* in the environment. The intermittent distribution of *B. pseudomallei* [49], combined with individual variation in exposure behavior, such as farming or gardening activities that involve contact with contaminated soil and water [50,51], may contribute to this discrepancy. Furthermore, host immune responses to *B. pseudomallei* are highly variable and can influence the detectability of antibody responses, with some individuals mounting strong cell-mediated immunity in the absence of measurable antibodies [52]. In addition, environmental sampling limitations, such as low sampling density or non-representative site selection may lead to underestimation of *B. pseudomallei* prevalence in soil [53]. These factors may contribute to the discordance between the environmental prevalence of *B. pseudomallei* and human seroprevalence.

This study has several limitations. First, our evaluation of culture-based methods was conducted using only a limited number of *B. pseudomallei* strains. Future studies should include a broader range of soil-dwelling bacteria to better assess the selectivity and overall microbial diversity influenced by each culture condition. In addition, the media optimization experiments were performed with sterile soil, which may not accurately reflect the microbial competition present in

natural environments. Future studies should utilize non-sterile soils during media optimization to better reflect natural ecological conditions. Second, the real-time PCR assays in our study were performed using direct soil samples, which may have reduced detection sensitivity due to the presence of PCR inhibitors or low DNA yield. In addition, the relatively small amount of soil used for DNA extraction (approximately 250 mg) may have further limited the sensitivity of real-time PCR, as *B. pseudomallei* is unevenly distributed in soil. Consequently, randomly selected sub-samples may not have contained sufficient bacterial cells for detection, thereby reducing assay sensitivity. This limitation could have affected the ability to detect *B. pseudomallei* in actual soil samples. Future studies should consider using prior enrichment or soil pellet for DNA extraction and incorporate validated internal controls to enhance the accuracy and reliability of real-time PCR assays. Finally, at some sampling locations and during certain periods, the soil was compacted or difficult to dig, which posed challenges for sample collection and may have limited the number of samples obtained. Therefore, sampling should ideally be conducted during the monsoon season, when soil conditions are more favorable. Addressing these limitations in future studies will improve the accuracy and applicability of environmental surveillance for *B. pseudomallei*.

Overall, this study highlights the effectiveness of culture-based and molecular methods for detecting *B. pseudomallei* from environmental soil samples. Our findings revealed a low prevalence of *B. pseudomallei* in Amnat Charoen Province. Moreover, this study provides additional insight into the environmental distribution and detection challenges of *B. pseudomallei*, and supports the integration of combined culture-based, molecular, and serological approaches for effective surveillance and development of public health responses for melioidosis.

## Supporting information

**S1 Table. Colony counts of eleven *B. pseudomallei* strains on Ashdown and ACER agar.** The colony forming unit (CFU/ml) of eleven *B. pseudomallei* was measured at 3 and 7 days of incubation at 37 °C. The experiments were performed in triplicate in three independent assays.
(DOCX)

**S1 Fig. Colony morphology of eleven *B. pseudomallei* strains on Ashdown and ACER agar.** The growth of *B. pseudomallei* was measured at 3 and 7 days of incubation at 37 °C in air. The experiments were performed in triplicate in three independent assays.
(PDF)

**S2 Table. Colony counts of eleven *B. pseudomallei* strains on TBSS-C50, TBSS-C50 based erythritol and ACER enrichment broth.** The log colony forming unit (log CFU/ml) of eleven *B. pseudomallei* was measured at 2, 5 and 9 days of incubation at 37 °C. The experiments were performed in triplicate in three independent assays.
(DOCX)

**S3 Table. Limit of detection (LOD) of real-time PCR targeting *BPSS1187* and *TTS1-orf2* genes in *B. pseudomallei* K96243-spiked PBS.** The table shows the limit of detection at nine serial dilutions ($10^8$-$10^0$ CFU/ml).
(DOCX)

**S4 Table. Linear regression analysis for real-time PCR targeting *BPSS1187* and *TTS1-orf2* genes.** The table shows the regression equations, correlation coefficients ($R^2$ values), and amplification efficiencies for each gene target.
(DOCX)

**S5 Table. Limit of detection (LOD) of real-time PCR targeting *BPSS1187* and *TTS1-orf2* genes in spiked soil with ten environmental *B. pseudomallei* strains.** The table shows the limit of detection at nine serial dilutions ($10^8$-$10^0$ CFU/ml).
(DOCX)

**S6 Table. Seroprevalence data of individuals in Amnat Charoen Province.** The table presents seroprevalence data, with the first cut-off value (0.418) highlighted in bold purple and the second cut-off value (1.165) highlighted in bold red. (DOCX)

**S7 Table. Detection of *B. pseudomallei* in soil samples using culture methods.** The table presents the detection results obtained from direct culture on Ashdown agar and enrichment culture in ACER broth at 5 and 9 days of incubation. (DOCX)

**S8 Table. Summary of *B. pseudomallei*-positive soil samples detected by culture and PCR methods in Amnat Charoen province.** Red color indicates samples that tested positive by each corresponding method. (DOCX)

## Acknowledgments

We would like to thank the staff at the Department of Microbiology and Immunology, Faculty of Tropical Medicine and Department of Medical Science, Amnat Charoen Campus, Mahidol University, for their assistance. We also thank all village health workers and field staff in Amnant Charoen Province for their support during soil collection.

## Author contributions

**Conceptualization:** Didtawa Suttisak, Narisara Chantratita.

**Data curation:** Didtawa Suttisak, Charlene Mae Salao Cagape, Rathanin Seng, Natnaree Saiprom, Wongwarut Boonyanugomol, Kamolchanok Rukseree, Narisara Chantratita.

**Formal analysis:** Didtawa Suttisak, Rathanin Seng, Paul J. Brett, Narisara Chantratita.

**Funding acquisition:** Narisara Chantratita.

**Investigation:** Didtawa Suttisak, Charlene Mae Salao Cagape, Rathanin Seng, Natnaree Saiprom, Wongwarut Boonyanugomol, Narisara Chantratita.

**Methodology:** Didtawa Suttisak, Paul J. Brett, Narisara Chantratita.

**Project administration:** Narisara Chantratita.

**Resources:** Paul J. Brett, Mary N. Burtnick, Narisara Chantratita.

**Software:** Narisara Chantratita.

**Supervision:** Narisara Chantratita.

**Validation:** Didtawa Suttisak, Narisara Chantratita.

**Visualization:** Didtawa Suttisak, Paul J. Brett, Narisara Chantratita.

**Writing – original draft:** Didtawa Suttisak, Charlene Mae Salao Cagape, Rathanin Seng, Mary N. Burtnick, Narisara Chantratita.

**Writing – review & editing:** Rathanin Seng, Wongwarut Boonyanugomol, Kamolchanok Rukseree, Paul J. Brett, Mary N. Burtnick, Narisara Chantratita.

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
