## [Decision Letter · Decision Letter 0]

25 Aug 2025

PNTD-D-25-01107

Evaluation of culture- and PCR-based methods for detecting Burkholderia pseudomallei in soil samples in Thailand

Dear Dr. Chantratita,

Thank you for submitting your manuscript to PLOS Neglected Tropical Diseases. After careful consideration, we feel that it has merit but does not fully meet PLOS Neglected Tropical Diseases's publication criteria as it currently stands. Therefore, we invite you to submit a revised version of the manuscript that addresses the points raised during the review process. Please pay attention to the comments provided by both reviewers who are the experts in the field of environmental surveys of Burkholderia pseudomallei. Please note that the review report from Reviewer 2 is provided in the attachment. 

Please submit your revised manuscript within 60 days Oct 24 2025 11:59PM. If you will need more time than this to complete your revisions, please reply to this message or contact the journal office at plosntds@plos.org. Please include the following items when submitting your revised manuscript:

We look forward to receiving your revised manuscript.

Kind regards,

Apichai Tuanyok, Ph.D.

Guest Editor

Elsio Wunder Jr

Section Editor

Shaden Kamhawi

co-Editor-in-Chief

Paul Brindley

co-Editor-in-Chief

**Additional Editor Comments (if provided):**

**Journal Requirements:**

**Reviewers' Comments:**

Reviewer's Responses to Questions

**Key Review Criteria Required for Acceptance?**

**Methods**

-Are the objectives of the study clearly articulated with a clear testable hypothesis stated?

-Is the study design appropriate to address the stated objectives?

-Is the population clearly described and appropriate for the hypothesis being tested?

-Is the sample size sufficient to ensure adequate power to address the hypothesis being tested?

-Were correct statistical analysis used to support conclusions?

-Are there concerns about ethical or regulatory requirements being met?

Reviewer #1: (No Response)

Reviewer #2: The global distribution of the soil pathogen B. pseudomallei is unclear. Among other reasons, this is also due to methodological challenges in detecting this pathogen in the environment. This manuscript addresses this important issue by validating previously described culture and PCR methods using spiked soil samples. In addition, culture and PCR methods were used to test the hypothesis that seropositivity in healthy individuals is associated with B. pseudomallei-positive soil samples of the respective households. The study design is appropriate to adress the stated objectives

**Results**

-Does the analysis presented match the analysis plan?

-Are the results clearly and completely presented?

-Are the figures (Tables, Images) of sufficient quality for clarity?

Reviewer #1: (No Response)

Reviewer #2: The results are generally clearly presented. Table and figures are of sufficient quality.

**Conclusions**

-Are the conclusions supported by the data presented?

-Are the limitations of analysis clearly described?

-Do the authors discuss how these data can be helpful to advance our understanding of the topic under study?

-Is public health relevance addressed?

Reviewer #1: (No Response)

Reviewer #2: The conclusions are generally justified. However the following point should be clarified and discussed:

Presumably, the soil samples for DNA extraction and for the cultures were separate sub-samples of the original 200 g samples? Given that the distribution of B. pseudomallei can vary greatly even within a small volume, the large differences in the soil quantities used for the PCR and culture protocols could at least partly explain the low PCR positivty rate of culture-positive samples (?).

**Editorial and Data Presentation Modifications?**

Reviewer #1: (No Response)

Reviewer #2: Line 490 Reference not correct?

Line 512 to 513 Meaning 20% higher among younger people than the average?

Line 556 to 561 The authors only refer to previous studies that detected B. pseudomallei DNA after culture enrichment. However, there are a number of studies which also applied PCR directly to soil extracts. These studies should be cited and briefly discussed.

Line 559 ...from soil ( ) Reference should be added.

Abstract line 39 to 43 The culture results of samples from seronegative and seropositive households should be provided

**Summary and General Comments**

Reviewer #1: (No Response)

Reviewer #2: (No Response)

PLOS authors have the option to publish the peer review history of their article (what does this mean? ). If published, this will include your full peer review and any attached files.

**Do you want your identity to be public for this peer review?** For information about this choice, including consent withdrawal, please see our Privacy Policy .

Reviewer #1: **Yes: ** Jeffrey Warner

Reviewer #2: No

**Figure resubmission:**
---

## [Decision Letter · Decision Letter 1]

8 Dec 2025

Dear Dr. Chantratita,

We are pleased to inform you that your manuscript 'Evaluation of culture- and PCR-based methods for detecting Burkholderia pseudomallei in soil samples in Thailand' has been provisionally accepted for publication in PLOS Neglected Tropical Diseases.

Best regards,

Apichai Tuanyok, Ph.D.

Guest Editor

Elsio Wunder Jr

Section Editor

Shaden Kamhawi

co-Editor-in-Chief

Paul Brindley

co-Editor-in-Chief

Reviewer's Responses to Questions

**Key Review Criteria Required for Acceptance?**

**Methods**

-Are the objectives of the study clearly articulated with a clear testable hypothesis stated?

-Is the study design appropriate to address the stated objectives?

-Is the population clearly described and appropriate for the hypothesis being tested?

-Is the sample size sufficient to ensure adequate power to address the hypothesis being tested?

-Were correct statistical analysis used to support conclusions?

-Are there concerns about ethical or regulatory requirements being met?

Reviewer #1: Objectives clear

Reviewer #2: (No Response)

**Results**

-Does the analysis presented match the analysis plan?

-Are the results clearly and completely presented?

-Are the figures (Tables, Images) of sufficient quality for clarity?

Reviewer #1: Results are complete

Reviewer #2: (No Response)

**Conclusions**

-Are the conclusions supported by the data presented?

-Are the limitations of analysis clearly described?

-Do the authors discuss how these data can be helpful to advance our understanding of the topic under study?

-Is public health relevance addressed?

Reviewer #1: Good conclusions

Reviewer #2: (No Response)

**Editorial and Data Presentation Modifications?**

Reviewer #1: (No Response)

Reviewer #2: (No Response)

**Summary and General Comments**

Reviewer #1: (No Response)

Reviewer #2: (No Response)

PLOS authors have the option to publish the peer review history of their article (what does this mean? ). If published, this will include your full peer review and any attached files.

**Do you want your identity to be public for this peer review?** For information about this choice, including consent withdrawal, please see our Privacy Policy .

Reviewer #1: **Yes: ** Jeff Warner

Reviewer #2: No

---

## [Editor Report · Acceptance letter]

Dear Dr. Chantratita,

We are delighted to inform you that your manuscript, "Evaluation of culture- and PCR-based methods for detecting Burkholderia pseudomallei in soil samples in Thailand," has been formally accepted for publication in PLOS Neglected Tropical Diseases.

Best regards,

Shaden Kamhawi

co-Editor-in-Chief

Paul Brindley

co-Editor-in-Chief
